

# **Simulating climate warming scenarios with intentionally biased**

# **bootstrapping and its implications for precipitation**

7   Taesam Lee

8   Dept. of Civil Engr., ERI, Gyeongsang National University,

9   501 Jinju-daero, Jinju, Gyeongnam, 660-701, South Korea

Corresponding Author: Taesam Lee, Ph.D.
Gyeongsang National University, Dept. of Civil Engineering
501 Jinju-daero, Jinju, Gyeongnam, 660-701, South Korea
Tel) +82-55-772-1797
Fax) + 82-55-772-1799
Email) tae3lee@gnu.ac.kr



29          **Abstract**

The outputs from GCMs provide useful information about the rate and magnitude of future climate
change. The temperature variable is the most reliable of the GCM outputs. However, hydrological
variables (e.g., precipitation) from GCM outputs for future climate change possess an uncertainty
that is too high for practical use. Therefore, a method, called intentionally biased bootstrapping
(IBB), that simulates the increase of the temperature variable by a certain level as ascertained from
observed global warming data is proposed. In addition, precipitation data was resampled by
employing a block-wise sampling technique associated with the temperature simulation. In
summary, a warming temperature scenario is simulated and the corresponding precipitation values
whose time indices are the same as the one of the simulated warming temperature scenario. The
proposed method was validated with annual precipitation data by truncating the recent years of the
record. The proposed model was also employed to assess the future changes in seasonal
precipitation in South Korea within a global warming scenario as well as in weekly time scale. The
results illustrate that the proposed method is a good alternative for assessing the variation of
hydrological variables such as precipitation under the warming condition.




## 1. Introduction


The complex influence of human actions on the climate system is well represented through global
climate models (GCMs). A number of GCMs demonstrate variations in the large-scale atmospheric
circulation and related changes in hydrometeorological variables (Allen and Ingram, 2002; Held
and Soden, 2006; Lenderink and Van Meijgaard, 2008). It has been generally accepted that to
quantify the range of possible changes in the hydrological cycle (such as precipitation and
evaporation) is harder than in temperature (Allen and Ingram, 2002). Furthermore, hydrological
variables vary much more in space and time than temperature and difficult to correctly simulate.
The relationship between temperature and precipitation has been studied in literature in order
to predict the future variations of precipitation under the global warming condition. From the
Clausius-Clapeyron (C-C) relation, saturation vapor pressure increases by 6-7% for each 1ºC
increase in temperature and rainfall intensity should also increase at the same rate with warming
(Trenberth and Shea, 2005). Lenderink and Van Meijgaard (2008) presented that 1hour intensity
of precipitation exhibit a C-C relation for summer while showing super C-C scaling for winter.
These relations are only focused on very short time scale (not more than daily) or generally
retrieved from GCM outputs. The behavior of mean precipitation over long-term period such as
months and seasons is difficult to predict as temperature increases. It might be beneficial if one
could derive the behavior of long-term mean precipitation under warming condition or the range
of possible changes.
Therefore, a simple method that simulates temperature from observed data is proposed in the
current study while increasing temperature up to a certain level as a warming scenario. In addition,
precipitation is simulated by employing a block-wise resampling technique (Srinivas and

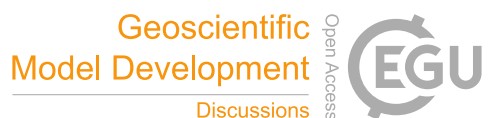

Srinivasan, 2000) associated with the temperature simulation. The resampled covariate,
precipitation, forcing the warming condition in a certain level is obtained from the simulation. The
proposed approach allows assessing the impact of precipitation as temperature increases with a
current climate horizon.

The paper is organized as follows. In the next section, the fundamental mathematical

background related to bias bootstrapping modeling is presented. The employed data and
application methodology are described in section 3. The validation study of the proposed IBB
approach is shown in section 4. The results assessing the long-term evolution of seasonal
precipitation with simulating weekly temperature and precipitation data are illustrated in section
5. Finally, the summary and conclusions are presented in section 6.

## 2. Methodology

In order to simulate warming scenario, i.e. increasing mean temperature, up to a certain level, the
observed data must be sampled with different combination.  Intuitively, warmer temperature
values are more likely to be resampled among the observations if the mean is increased. Therefore,
the proposed method in the current study is to resample the observed data by fixing the mean
temperature increment in the resampled dataset by weighting the probability of selection according
to its magnitude (see Figure 1).  In addition, the block bootstrapping with precipitation was
employed to assess the changes in these variables as temperature increases.

### 2.1. Intentionally Biased Bootstrapping (IBB)

Bootstrapping (also known as resampling from observed data with replacement) is a statistical
method for creating replica datasets from the original data to assess the variability of the quantities
of interest without analytical calculation (Davison and Hinkley, 1997; Davison et al., 2003; Ouarda



and Ashkar, 1995). This bootstrapping technique has been extended to simulate time series of hydrometeorological variables (Beersma and Buishand, 2003; Lall et al., 1996; Lall and Sharma, 1996; Lee and Ouarda, 2011, 2010; Mehrotra and Sharma, 2005). In the current study, the intentionally bias bootstrapping (**IBB**) technique is employed so that the mean of the resampled datasets are varied as needed to simulate a global warming scenario.

IBB was proposed by Hall and Presnell (1999) as a class of weighted bootstrapping techniques in order to reduce bias or variance as well as to render some characteristic equal to a predetermined quantity. A good example of IBB is the adjustment of Nadaraya-Watson kernel estimators to make them competitive with local linear smoothing(Cai, 2001). In the current study, IBB was employed to simulate the temperature data from observation by bootstrapping under the constraint of increasing mean value, which indicates warming. The conceptual background of IBB has been employed to simulate future climates of weather analogs (Orlowsky et al., 2010; Orlowsky et al., 2008). In the current study, a IBB method with easy manipulation to simulate increased temperature data is proposed. The mathematical description of the proposed IBB method is the following.

Among an $n$ number of observations $x_i$ , where $i=1,…,n$, assume resampling the observations with replacement (i.e. bootstrapping) by increasing the mean of the simulated data by as much as $\Delta_\mu$ ; this implies that higher values have a higher probability of being resampled and lower values have lower selection probability. This IBB can be achieved by assigning different weights $S_{i,n}$ according to the magnitudes of the observations as

$$S_{i,n} = i/n \tag{1}$$





Note that this assigned weight $S_{i,n}$ plays a role in the selection probability for the observed data in
the IBB procedure after scaling and adjusting it.
The mean of the resampled data is
$$\tilde{\mu} = \frac{1}{\Psi} \sum_{i=1}^{n} S_{i,n} x_{(i)} \qquad (2)$$

where $x_{(i)}$ represents the $i^{\text{th}}$ increasing ordered value and $\Psi = \sum_{i=1}^{n} S_{i,n}$. The amount of the mean
increase $\delta_{\mu}$ is
$$\delta_{\mu} = \tilde{\mu} - \hat{\mu} = \frac{1}{\Psi} \sum_{i=1}^{n} S_{i,n} x_{(i)} - \frac{1}{n} \sum_{i=1}^{n} x_i \qquad (3)$$

To obtain different values of $\delta_{\mu}$, the weights can be generalized with the weight order ($r$) as
$$\tilde{\mu}(r) = \frac{1}{\Psi_r} \sum_{i=1}^{n} S_{i,n}^r x_{(i)} \qquad (4)$$

where $\Psi_r = \sum_{i=1}^{n} S_{i,n}^r$. The difference is
$$\delta_{\mu}(r) = \tilde{\mu}(r) \quad \hat{\mu} = \frac{1}{\Psi_r} \sum_{j=1}^{n} S_{j,n}^r x_{(j)} - \frac{1}{n} \sum_{j=1}^{n} x_j \qquad (5)$$

Once the magnitude of the mean increase is given (e.g., temperature increase) as $\Delta_{\mu}$, the weight
order '$r$' is estimated accordingly. For example, when the temperature change is obtained from
the GCM outputs and this change is supposed to be propagated into a specific location and a finer



time scale, the selection of the weight order can be performed using a meta-heuristic optimization
technique with the objective function as

$$\text{Minimize } [\Delta_\mu - \delta_\mu(r)]^2 \tag{6}$$

In the current study, the harmony search (HS) was used for the meta-heuristic optimization. The
performance of the HS in hydrological applications is well reviewed in the literature (Geem et al.,
2001; Lee and Geem, 2005, 2004; Lee and Jeong, 2014a; Mahdavi et al., 2007; Yoon et al., 2013a).
Note that if $r > 0$, then $\delta_\mu(r) > 0$, which implies a global warming scenario; if $r < 0$, then
$\delta_\mu(r) < 0$, which implies a global cooling scenario. When $r < 0$, lower values are resampled more
frequently than are higher values. causing the mean of the resampled data to decrease. Furthermore,
if $r$ goes to infinity then the maximum of the observations is always selected, and if $r$ goes to
negative infinity, only the minimum is chosen.

In the IBB procedure, the adjusted scaled weight $\eta_i = S_{i,n}^r / \Psi_r$ is the probability that each $i^{\text{th}}$

data point is subject to be selected. In the case of $n=30$, the weights for $i=1,\dots,n$ are shown in
Figure 2 with the weight order of $r=0.5$. The figure presents that the probability of being selected
(i.e., $\eta_i$) is between approximately 0.01 for the lowest values and 0.05 for the highest order values
of approximately 0.05 to lead positive bias in the resampled data (e.g., 1.0℃ increase). For
example, if the number of the simulation is 100 and $\eta_i$=0.05, then the data point will be selected
5 times. A different probability implies a different number of selection for each data point.
Subsequently, a different number of selections may lead to variation changes, called variance
reduction or inflation. This issue is dealt with in the following section.

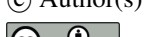

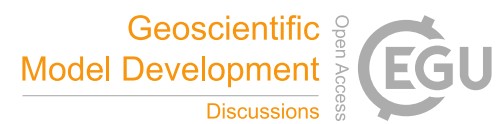

**2.2. Variance reduction and inflation**
Because of the biased selection of higher values, the variance of the resampled data results is
reduced (Lee and Jeong, 2014a; Lee and Ouarda, 2010; Lee et al., 2010a; Salas and Lee, 2010;
Sharif and Burn, 2006). The estimated variance of the simulated data with IBB is

$$\tilde{\sigma}^2(r) = \sum_{j=1}^{n} \frac{S_{j,n}^r}{\Psi_r} x_{(j)}^2 - \tilde{\mu}^2 \qquad (7)$$

Note that the variance in Eq. (7) is based on $\sigma^2 = E(X^2) - (EX)^2$. The difference of the variance
is

$$\delta_{\sigma^2}(r) = \hat{\sigma}^2 - \tilde{\sigma}^2(r) \qquad (8)$$

where $\hat{\sigma}^2$ is the sample variance of the observed data. To overcome the reduction of the variance
in IBB, a random perturbation can be applied to the resampled data $X_R$ as

$$X_R* = X_R + \sqrt{\delta_{\sigma^2}(r)}\varepsilon \qquad (9)$$

where $\varepsilon$ is a random variable with a normal distribution $N(0,1)$. Subsequently, the mean and
variance of the perturbed data are

$$\hat{\mu}_{R*} = \tilde{\mu} \qquad (10)$$

$$\hat{\sigma}_{R*}^2 = \tilde{\sigma}^2 + \delta_{\sigma^2}(r) = \tilde{\sigma}^2 + \hat{\sigma}^2 - \tilde{\sigma}^2(r) = \hat{\sigma}^2 \qquad (11)$$

**2.3. Block bootstrapping**
When the temperature presumably increases by a certain degree, it is interesting to note how the
other weather variables vary. For example, if the temperature is increased by 1°C, the greatest



concern in climate research will be how the precipitation will change. To address this question,
the block bootstrapping technique for the precipitation variable is adapted (Carlstein et al., 1998;
Lee et al., 2010b). Once the temperature is resampled from the observed data at certain times using
IBB, the observed precipitation data from the same time are considered (see Figure 2). Unlike for
the case of temperature, there is no variance reduction in the resampled precipitation data because
the precipitation data are not conditionally resampled. This block bootstrapping technique is
popularly employed in multivariate weather simulations (Lee and Jeong, 2014b; Lee et al., 2012)..

### 170    2.4. Overall Simulation Procedure

The overall simulation procedure of temperature and precipitation data is described in this section.
Simple schematic presentation of the procedure is shown in Figure 1.
Let $x_i$, $y_i$ ($i=1,\ldots,n$) be the observed temperature and precipitation data, respectively. Suppose that
the simulation length is the same as the record length (i.e. $n$) and100 series need to be simulated.
(a) Assume that the increased overall temperature mean is known as $\Delta_\mu$.
(b) Estimate the weight order (r) from meta-heuristic algorithm (here, Harmony Search) with

the objective function of Eq.(6) from the observed temperature data.

(c) Resample the temperature data from the observations with the probability of $S_{i,n}^r$ for $i^{\text{th}}$

largest data ($i=1,\ldots,n$).

(d) Assume that $k^{\text{th}}$ largest temperature data $x_{(k)}$ is resampled from step (3) and its

corresponding time index of ($k$) is '$j$'. Note that ($k$) indicates the $k^{\text{th}}$ largest value and j





indicates the $j^{th}$ time-index value. Then, $j^{th}$ precipitation data, $y_j$, is resampled

simultaneously.

(e) Apply Eq.(9) to the resampled temperature data from step(3) (say, $x_{(k)} + \sqrt{\delta_{\sigma^2}(r)}\varepsilon$ ), if the

variance inflation is chosen.

Note that the current procedure is explained for the case of no seasonal variability due to

simplicity. In other words, the explained procedure above must be applied at each week or each
month for weekly or monthly data. The detailed description of the proposed method for the case
of monthly precipitation data with the full record is provided in the supplementary material
(Supplement A).

## 3. Data description and application methodology


In the current study, weather stations that record temperature and precipitation in South

Korea (74 locations) and that are managed by the Korea Meteorological Administration (KMA)
were employed. South Korea is located in Far East Asia and has a mean annual precipitation of
1283 mm. This country is climatologically influenced by the Siberian air mass during winter and
the Maritime Pacific High during summer. Most of the annual precipitation in South Korea falls
during the rainy season from June to September due to the occurrence of tropical cyclones,
extratropical cyclones, fronts and other weather systems. Because the orographic area in South
Korea is heterogeneous and large, the rainfall in South Korea has large spatial and temporal
variability (Park et al., 2007; Yoon et al., 2013b). The water resource control system, including
climate change, is an important aspect of this study due to the seasonal and spatial variability of
rainfall in this country.



Datasets shorter than 30 years of data were excluded, after which a total of 54 datasets were
employed. The data were extracted from the KMA website (http://www.kma.go.kr/). Most of the
time spans are approximately 33 years, from 1976 to 2008.
The validation study was performed with annual dataset to present the performance of the
proposed model with truncating recent years as 1994-2008. The truncated data was not used in
simulation but employed in comparison. Also, a case study was applied with the weekly dataset of
the 54 stations in South Korea. In the application study of the proposed IBB procedure in section
5, (1) 0.5°C and 1.0°C increases in the mean weekly temperature were assumed; (2) weekly
temperature datasets were simulated using the assumed temperature increase; (3) weekly
precipitation datasets were also simulated along with the weekly temperature dataset as a block.
Note that the simulation does include not a gradual change, such as a trend, but the overall mean
change. We simulated the weekly time scale so that the data spanned a long enough period to
provide a summary of weather statistics and a short enough period to reflect the temporal
variability. Furthermore, the observed weekly datasets of temperature and precipitation were
aggregated into seasonal time scale data, and the aggregated seasonal data were used to present
the seasonal variations in precipitation as temperature increases.
Note that although we simulated the temperature with a specific condition of increase (e.g.
+0.5 °C or +1.0°C), no such restriction was placed on the precipitation, allowing one to determine
whether there is any change in precipitation with the condition of increasing temperature. One
hundred series were simulated with the same time span as the observations.



## 4. Validating IBB model with annual data

To further obtain the credibility of the proposed IBB model, we validated the model with truncating
the last 15 years (1994-2008) of the annual mean temperature and precipitation data over South
Korea. The last truncated 15 years were set as the validation period while the rest of the preceding
years as the test period. The dataset of the test period was employed in simulation while the dataset
of the validation period is only used in comparison to check how much the proposed model
performs. Among others, annual scale data is employed to easily illustrate the performance of the
proposed IBB model. At first, some mathematical terms need to be defined to explain the
validation procedure as follows.

$$D\mu_p^{obs} = \mu p_V - \mu p_T \tag{12}$$

$$D\mu_p^{IBB} = \mu p_{IBB} - \mu p_T \tag{13}$$

where $\mu p_V$ and $\mu p_T$ are the mean annual precipitation over the validation years and over the test
period, respectively, while $\mu p_{IBB}$ is the annual mean precipitation of the IBB simulated data with
the record length of the validation years. The same denotation as the precipitation variable is taken
for the temperature variable as $\mu T_V$, $\mu T_T$, $\mu T_{IBB}$, $D\mu_T^{obs}$, and $D\mu_T^{IBB}$.
The validation procedure is (1) to truncate the 15 years (1994-2008) of annual temperature
and precipitation for each station; (2) to estimate the mean differences of the annual temperature
and precipitation between the validation period (1994-2008) and the test period (1976-1993),
$D\mu_T^{obs}$ and $D\mu_p^{obs}$, respectively; (3) to perform the IBB simulation with the annual precipitation and
temperature of the test period conditioned on the estimated mean differences of the temperature



between two periods (i.e. $D\mu_T^{obs}$) for each station; and (4)   to compare the estimated mean
differences of the observed precipitation (i.e. $D\mu_p^{obs}$) with the mean differences between the IBB
simulated precipitation and the precipitation for the test period (i.e. $D\mu_p^{IBB}$).

The annual mean temperature differences between the validation period and the test period

at each station is presented in Figure 3 for the IBB simulated data ($D\mu_T^{IBB}$, boxplot) and the
observed data ($D\mu_T^{obs}$, circle). The figure indicates that the IBB model fairly well simulates the
temperature data as much as it was intended, except few stations that shows high increase
especially with more than one-degree increase (e.g. stations 6 and 7). Note that the employed test
period is relatively short and not enough number of high values of annual temperature is included
during the test period and this might result the underestimation of the intended temperature
increase.

In Figure 4, the annual mean precipitation of the observation over the validation period ($\mu p_V$,

filled blue circle) and the test period ($\mu p_T$, filled red triangle) as well as the IBB simulation ($\mu p_{IBB}$,
boxplot) is illustrated. The result indicates that the observed mean precipitation over the validation
period ($\mu p_V$) presents higher than the mean for the test period ($\mu p_T$) in most of the stations. The
IBB simulated data reflects this tendency showing higher mean precipitation than the mean
precipitation of the test period though its magnitude shows some difference.

The mean of the observed annual precipitation for the validation period at each station and

the mean of one hundred IBB simulated data is presented in Figure 5. The top panel presents that
the simulated data fairly well reproduce the observed mean of annual precipitation for the
validation period (1994-2008). The observed mean difference ($D\mu_p^{obs}$) of the annual precipitation



between the test period (1976-1993) and the validation period shown at the bottom panel of Figure
5 fairly matches with the one of the IBB simulated data ($D\mu_p^{IBB}$). Rather high variability at the
difference is inevitable due to relatively small record length for both the test period and the
validation period. Overall, the validation study implicates that the proposed IBB approach can
simulate the future evolution of annual precipitation over South Korea.

In Figure 6, the spatial distribution of the differences for the annual mean precipitation is

presented with the observed data (i.e. $D\mu_p^{obs}$) and with the IBB simulated data ($D\mu_p^{IBB}$). High
increase of annual mean precipitation in the north and south part of the country and small increase
and slight decrease in the south part shown in the observed data (left panel) is well reflected in the
IBB simulated data (right panel) except that the increase is shown from the IBB simulated data
(right panel) in the left south part of the country is not shown in the observed data. Overall, the
figure indicates that the spatial pattern of the annual mean precipitation difference from the
observed data (see the left panel) is similar to the one from the IBB simulated data (see the right
panel).
**5. Precipitation changes according to assumed temperature increase**
Figure 7 shows the results of the fitted IBB model for the Buan station, located at 35º 44' N and
126º 43' E. The top panel (Figure 7(a)) shows the estimated weight order of each week for the
mean temperature data employing the HS meta-heuristic algorithm with the objective function of
Eq. (6) while assuming a 0.5°C increase. The estimated values range from 0.2 to 1.3. The mean
and standard deviation of the observed and theoretical results (see Eqs. (2) and (7)) with a 0.5°C
mean increase are shown in Figure 7(b) and (c), respectively. The predominant annual cycle of the





mean weekly temperature is seen in the mean statistics, as shown in Figure 7(b), while the annual
cycle of the standard deviation (equivalent to the square root of variance) is not as prominent as
the annual cycle of the mean (see Figure 7(c)). Note that the weight order and the standard
deviation (see Figure 7(a) and (c)) are highly negatively correlated. In other words, when the
standard deviation is small (e.g., at approximately the 23$^{rd}$ week), the weight order is high and vice
versa. This result is intuitive in that if the variance is great, the corresponding temperature values
differ greatly from each other. Subsequently, the weights of the large values to be selected are not
necessarily much different from the weights of the low values in such a case, which induces a low
weight order. In Figure 7(c), the variance difference between the observed and theoretical data, as
defined in Eq. (8), is shown with a dotted line. This variance difference is inflated to the resampled
data, as in Eq. (9). This inflation procedure is optional in assessing the overall trend of annual
mean precipitation data regarding climate warming scenarios. However, it might be helpful when
the purpose of the study is to evaluate an overall variation of extreme precipitation statistics.

The statistics of the simulated data from IBB with the condition of a 0.5℃ degree mean

temperature increase are shown as a boxplot in Figure 8; the statistics of the observed data are
shown in the same figure with dotted lines and cross marks. The mean increases by exactly 0.5℃,
as intended, and the standard deviation (square root of variance) is well preserved through the
variance inflation process (see Eq. (8)). The minima and maxima of the mean weekly temperatures
are increased.

Shown in Figure 9(a) are the mean differences between the simulated and observed weekly

precipitation with the conditions of 0.5℃ and 1.0℃ increases at the Buan station. The differences



are not significant at the 5% level. However, the mean differences are continuously positive from
the 30th to 40th week, which is during the summer season. This result indicates that a seasonal
effect on the precipitation change must exist. Therefore, we also extended our study to a seasonal
time scale. The mean precipitation differences of all 54 stations are shown for 0.5℃ and 1.0℃
increases in Figure 9(b) and (c), respectively. Both plots show a decrease in autumn and increases
in the other seasons.
For a 1.0℃ temperature increase, 61%, 24%, and 45% of the employed stations show a
significant increase in mean precipitation for the winter, spring, and summer seasons, respectively.
In contrast, the mean temperature decreases during the autumn season. Approximately 30% of the
stations experience a significant change in the mean precipitation at the 5% level given a 1.0℃
temperature increase. The detailed information is provided in Table 1.
The spatial distribution of seasonal mean precipitation differences is presented in Figure 10
given the condition of a 1℃ temperature increase. An increasing pattern of precipitation during
winter (see Figure 10(a)) can be seen over the South Korea peninsula. Notably, the eastern and
southern coastal areas undergo a significant increase with a 95% confidence interval (±5.38). Note
that the significance interval at each station is different because the variances between stations are
different. The detailed significance interval for each station is provided in Table 2. During spring
(see Figure 10(b)), the northern part of the country shows an increasing pattern while the
southwestern and southeastern parts show decreasing patterns, but their magnitudes are not
significant (±15.04). The summer precipitation (see Figure 10(c)) undergoes a significant increase
in the southwest area of the country (±29.94). In contrast to the other seasons, a significant decrease



in mean precipitation occurs during autumn (see Figure 10(d)) throughout the country, especially
over the eastern coastal area. The same spatial pattern of seasonal mean precipitation can be
observed given the condition of a 0.5°C temperature increase, as in the case of a 1.0°C temperature
increase, with little significant change (see Figure 11).

The spatial distributions of seasonal precipitation changes seem to be related to the flow

direction of the seasonal air mass. In South Korea, winter is influenced primarily by the Siberian
air mass with prevailing northwesterly winds, while summer is hot and humid with southeasterly
winds.

## 6. Summary and Conclusions

A simple method is proposed (1) to simulate precipitation given the condition of a mean
temperature increase derived from the observations and (2) to address the problem of how the
precipitation vary while the temperature is increased through global warming. The results
illustrated that a simple IBB technique for the temperature variable incorporating block sampling
of precipitation can achieve this objective.

The presented technique is valuable because hydrometeorological variables such as

precipitation and discharge are difficult to model with current GCMs, while the temperature
prediction is relatively accurate. The proposed method can be extended to other
hydrometeorological variables as well as other applications, including studies at the global scale.
The limit of the proposed method is that the temperature increase is limited since employed data
is observational. One possibility for allowing a greater temperature increase than that from the
observations is to include neighboring, similar stations or seasons. The author believes that the



proposed model can be a good surrogate or competitor in GCM-based climate change impact
assessments of hydrometeorological variables.

## 7. Code and Data Availability

All the employed code can be provided upon the request to the author of the current study. The
employed precipitation and temperature data over South Korea can be downloaded from the KMA
website http://www.kma.go.kr/weather/climate/past_cal.jsp .

## Acknowledgements

This work was supported by the National Research Foundation of Korea (NRF) grant funded by
the Korean government (MEST) (No. 2015R1A1A1A05001007). All the employed data can be
provided upon the request to the author of the current study.




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



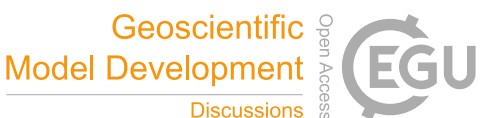

Table 1. Mean precipitation difference of the observed and simulated data for seasonal data over

all the employed stations in South Korea in case of +1.0 °C mean temperature increase.

| Station | Mean Diff | | | | Station | Mean Diff | | | |
|---|---|---|---|---|---|---|---|---|---|
| | Winter | Spring | Summer | Autumn | | Winter | Spring | Summer | Autumn |
| 1 | **11.2** | **14.3** | 20.2 | -12.0 | 28 | 2.6 | 12.1 | 9.6 | -4.1 |
| 2 | 3.2 | **22.4** | 4.5 | 0.0 | 29 | **4.4** | **20.6** | **50.4** | -3.8 |
| 3 | 11.0 | 5.0 | 21.5 | **-17.2** | 30 | **5.5** | 11.7 | 30.0 | -4.2 |
| 4 | 1.6 | **15.7** | **38.1** | -2.3 | 31 | **4.4** | **19.2** | 15.8 | -4.7 |
| 5 | 1.5 | 11.9 | 3.9 | -6.2 | 32 | 4.2 | **15.9** | 18.0 | -2.0 |
| 6 | 1.7 | 10.1 | **28.5** | -2.0 | 33 | **6.6** | **16.4** | **46.1** | -4.2 |
| 7 | 1.7 | 8.2 | 16.8 | -2.3 | 34 | **9.5** | 9.5 | **32.6** | **-7.1** |
| 8 | 3.2 | **22.3** | 33.6 | -3.1 | 35 | **6.4** | 1.7 | **44.1** | -6.8 |
| 9 | 2.3 | **19.1** | **15.0** | -4.9 | 36 | 5.1 | -4.2 | **52.1** | **-9.4** |
| 10 | **9.8** | 6.7 | 21.4 | **-16.3** | 37 | 5.6 | 7.4 | **39.9** | **-9.4** |
| 11 | 2.8 | **18.8** | **30.3** | -3.3 | 38 | **9.2** | -4.3 | **53.8** | -3.1 |
| 12 | 5.3 | 10.8 | **32.9** | -7.2 | 39 | **9.6** | -3.2 | **65.0** | -5.6 |
| 13 | **5.1** | 3.5 | 21.5 | **-9.3** | 40 | **11.5** | -9.9 | **82.2** | -6.5 |
| 14 | **9.8** | 1.2 | 28.8 | -4.5 | 41 | **9.1** | 4.2 | **33.3** | **-7.4** |
| 15 | **6.6** | -0.9 | 11.5 | **-5.1** | 42 | **9.6** | -11.5 | **61.2** | **-8.1** |
| 16 | **5.9** | -1.0 | **32.6** | **-7.5** | 43 | 4.2 | 12.9 | **42.7** | -3.0 |
| 17 | **10.2** | -9.3 | 26.7 | 0.6 | 44 | **6.3** | **20.2** | 33.8 | -2.6 |
| 18 | **8.2** | -1.7 | **50.2** | -4.5 | 45 | **12.9** | 8.8 | 10.5 | **-7.9** |
| 19 | **13.2** | -2.7 | 23.4 | 0.8 | 46 | **5.8** | **11.2** | 19.4 | -3.8 |
| 20 | **9.8** | -4.3 | 33.1 | -0.7 | 47 | 3.1 | **14.3** | **56.3** | **-7.0** |
| 21 | **8.1** | -15.4 | 12.4 | -4.5 | 48 | **7.1** | -2.4 | 14.8 | **-4.7** |
| 22 | **7.8** | -6.0 | **52.3** | -2.3 | 49 | **9.0** | 3.4 | **68.4** | **-5.9** |
| 23 | **11.4** | -17.5 | 19.7 | **-12.6** | 50 | 4.2 | 2.1 | 31.6 | -2.3 |
| 24 | 1.9 | 11.2 | 21.1 | 0.1 | 51 | **8.9** | 5.5 | **39.5** | -3.2 |
| 25 | 2.3 | 8.6 | 21.8 | -2.4 | 52 | **8.6** | 8.0 | **78.2** | -1.5 |
| 26 | 2.3 | 8.8 | 13.4 | 0.8 | 53 | **16.4** | 6.0 | 28.8 | -4.1 |
| 27 | 2.5 | 9.3 | 26.0 | -2.9 | 54 | **10.5** | 20.9 | 23.2 | 1.7 |
| | Mean confidence interval | | | | | ±5.38 | ±15.04 | ±29.94 | ±7.01 |
| | # of Significant Stations | | | | | 33 | 13 | 25 | 16 |
| | (percent) | | | | | (61%) | (24%) | (46%) | (30%) |






Table 2. Confidence interval for mean precipitation difference of the observed and simulated data
for seasonal data.

| Station | Winter | Spring | Summer | Autumn | Station | Winter | Spring | Summer | Autumn |
|---|---|---|---|---|---|---|---|---|---|
| 1 | 10.7 | 12.4 | 28.4 | 13.6 | 28 | 3.89 | 14.15 | 32.45 | 6.08 |
| 2 | 3.7 | 13.2 | 29.0 | 5.1 | 29 | 4.71 | 14.76 | 31.49 | 6.34 |
| 3 | 12.7 | 10.3 | 29.6 | 14.2 | 30 | 5.24 | 14.79 | 30.39 | 5.55 |
| 4 | 3.7 | 14.6 | 34.7 | 7.6 | 31 | 4.08 | 14.26 | 27.61 | 7.45 |
| 5 | 3.6 | 12.0 | 25.9 | 7.8 | 32 | 4.25 | 14.31 | 28.31 | 7.09 |
| 6 | 4.0 | 12.0 | 25.3 | 5.6 | 33 | 5.00 | 15.87 | 31.29 | 8.08 |
| 7 | 3.6 | 14.0 | 25.9 | 7.7 | 34 | 5.62 | 13.73 | 25.75 | 6.06 |
| 8 | 4.1 | 13.7 | 26.4 | 6.4 | 35 | 4.86 | 12.44 | 30.64 | 6.93 |
| 9 | 4.1 | 14.8 | 27.1 | 8.6 | 36 | 5.61 | 12.53 | 27.52 | 7.52 |
| 10 | 8.9 | 10.5 | 26.7 | 11.4 | 37 | 5.32 | 12.89 | 26.21 | 7.28 |
| 11 | 4.8 | 14.5 | 23.0 | 7.0 | 38 | 5.12 | 13.53 | 32.37 | 5.46 |
| 12 | 5.5 | 15.2 | 30.7 | 6.4 | 39 | 5.15 | 15.64 | 34.46 | 6.45 |
| 13 | 4.6 | 13.1 | 24.6 | 5.2 | 40 | 5.27 | 20.28 | 37.15 | 6.87 |
| 14 | 8.2 | 12.9 | 30.9 | 6.7 | 41 | 4.80 | 20.76 | 29.50 | 5.57 |
| 15 | 4.8 | 12.1 | 23.6 | 4.5 | 42 | 5.20 | 21.00 | 35.75 | 7.88 |
| 16 | 5.6 | 12.5 | 26.9 | 6.3 | 43 | 4.45 | 15.73 | 26.47 | 6.16 |
| 17 | 7.2 | 15.7 | 30.1 | 6.9 | 44 | 5.23 | 14.63 | 26.25 | 5.11 |
| 18 | 5.2 | 15.4 | 31.9 | 5.7 | 45 | 8.23 | 11.25 | 24.05 | 7.16 |
| 19 | 6.9 | 20.1 | 35.1 | 8.7 | 46 | 4.30 | 10.81 | 24.10 | 4.29 |
| 20 | 6.0 | 19.3 | 34.3 | 7.5 | 47 | 4.60 | 11.30 | 25.36 | 4.91 |
| 21 | 4.6 | 15.7 | 26.5 | 6.1 | 48 | 4.80 | 11.24 | 23.40 | 4.32 |
| 22 | 5.0 | 19.5 | 30.1 | 6.9 | 49 | 5.81 | 12.41 | 34.88 | 5.73 |
| 23 | 5.4 | 22.6 | 39.4 | 8.4 | 50 | 5.38 | 14.71 | 33.37 | 5.54 |
| 24 | 3.6 | 17.3 | 27.5 | 8.3 | 51 | 4.73 | 15.29 | 30.09 | 6.00 |
| 25 | 3.6 | 13.1 | 30.8 | 6.6 | 52 | 6.32 | 17.35 | 41.62 | 7.15 |
| 26 | 4.0 | 13.5 | 28.2 | 6.9 | 53 | 7.70 | 29.41 | 44.00 | 11.16 |
| 27 | 3.3 | 13.5 | 27.7 | 4.6 | 54 | 7.56 | 23.95 | 42.12 | 9.89 |







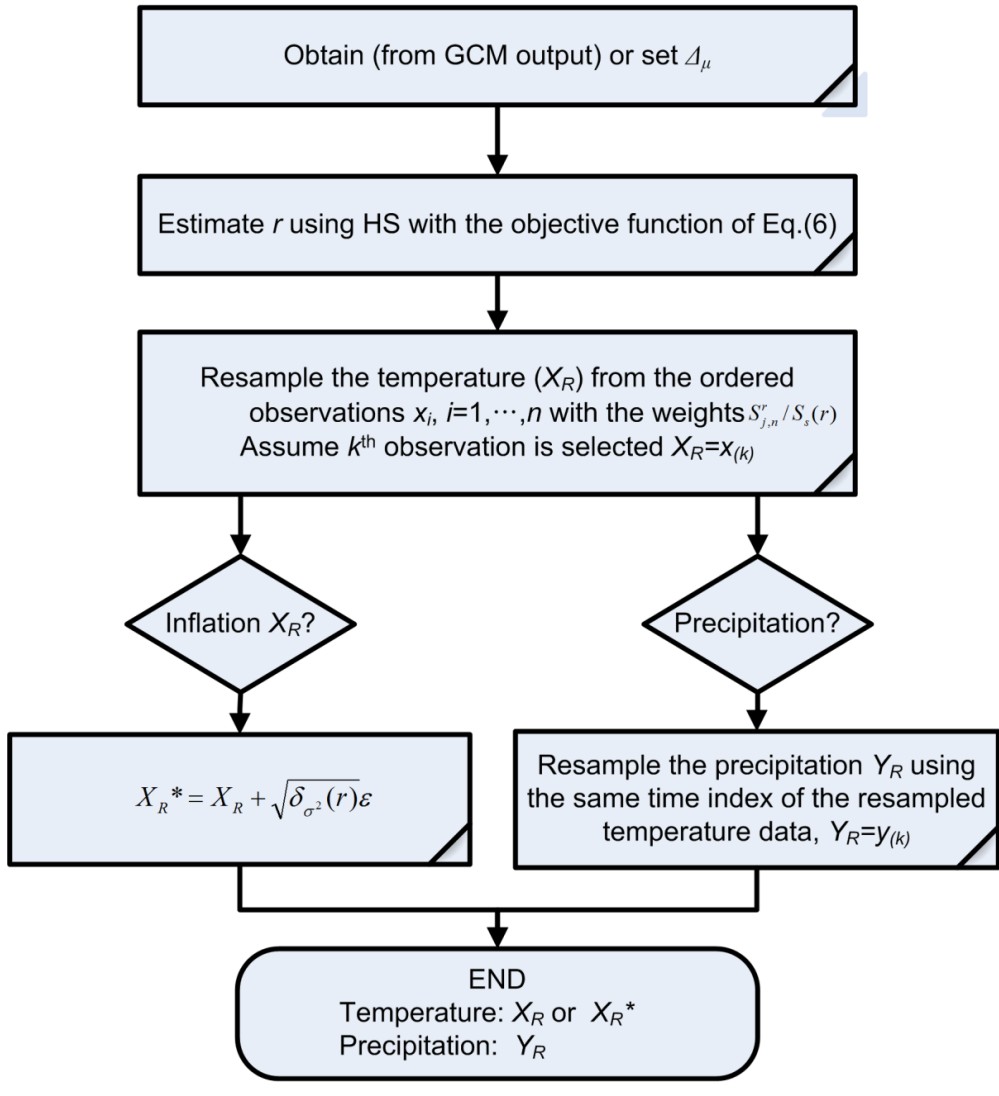


Figure 1. Procedure for the proposed simulation IBB method of temperature and precipitation data.





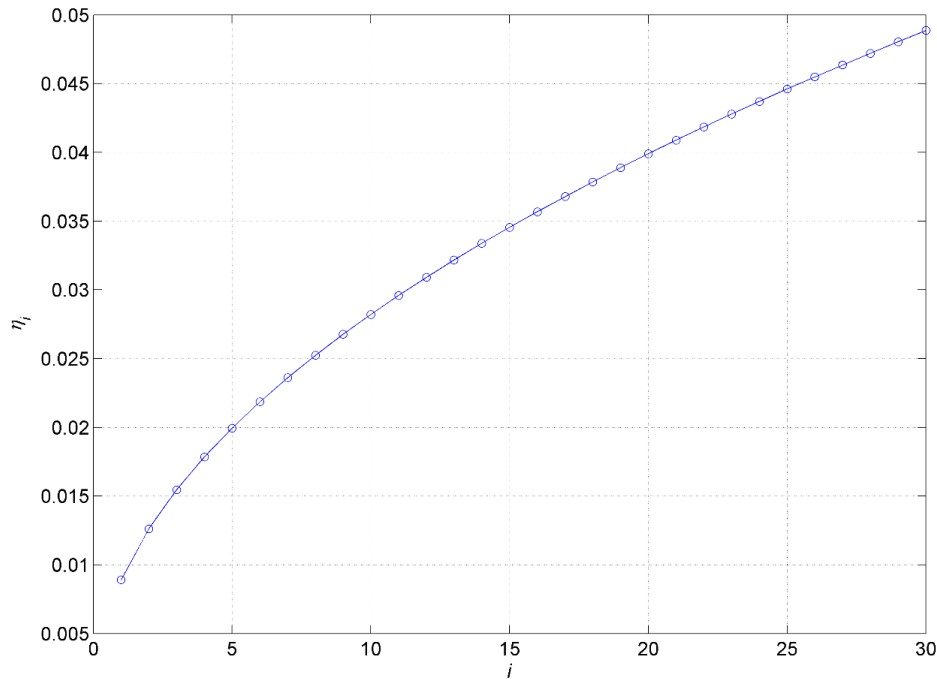


Figure 2. Example of the adjusted scaled weights ($\eta_i$) vs. order numbers in the case of $n$=30 and
order weight $r$=0.5. Note that $\eta_i$ is the probability of being selected and increases as the order is
increased, so that higher values are subject to being selected more often than are lower values,
leading to a positive bias.




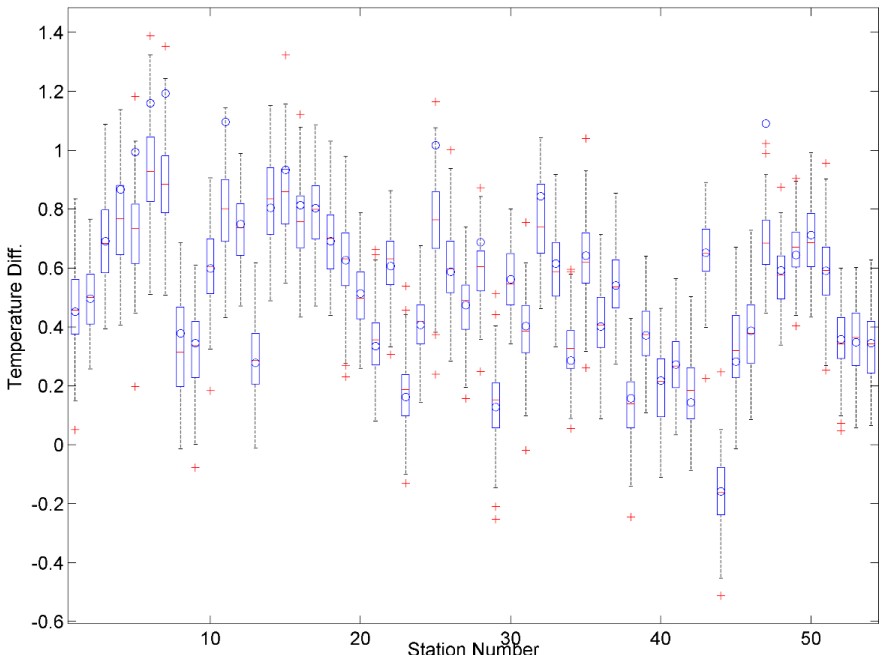

Figure 3. Annual mean temperature difference between the validation period (1994-2008) and
the test period (1976-1993) for each station for the IBB simulated data (boxplot) and the
observed data (circle). Boxes indicate the interquartile range (IQR), and whiskers extend to +/-
1.5IQR. The horizontal lines inside the boxes depict the median of the data. Data beyond the
fences (+/-1.5IQR) are indicated by a plus symbol (+), which represent outliers.





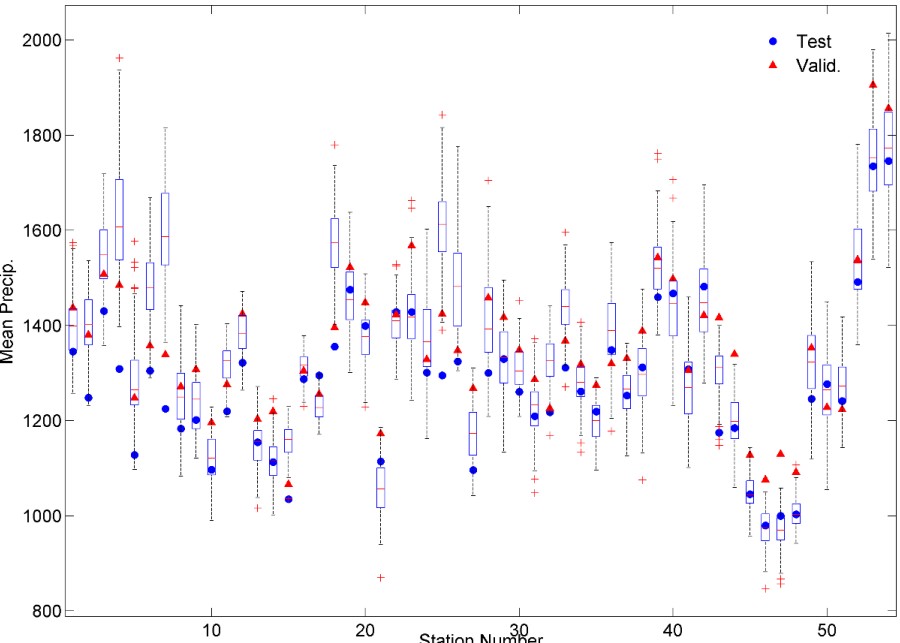


Figure 4. Annual mean precipitation of the IBB simulation (boxplot) and the observation over the
validation period (filled blue circle) as well as the test period (filled red triangle) conditioned with
the temperature change (see Figure 3). Note that the observed mean precipitation over the
validation period (1994-2008) (see the red triangles) shows mostly higher than the mean over the
test period (1976-1993) (see the blue circles). Also, the IBB simulated precipitation (boxplot)
reflects this tendency showing higher than the mean precipitation of the test period (blue circles).
Boxes indicate the interquartile range (IQR), and whiskers extend to +/-1.5IQR. The horizontal
lines inside the boxes depict the median of the data. Data beyond the fences (+/-1.5IQR) are
indicated by a plus symbol (+), which represent outliers.




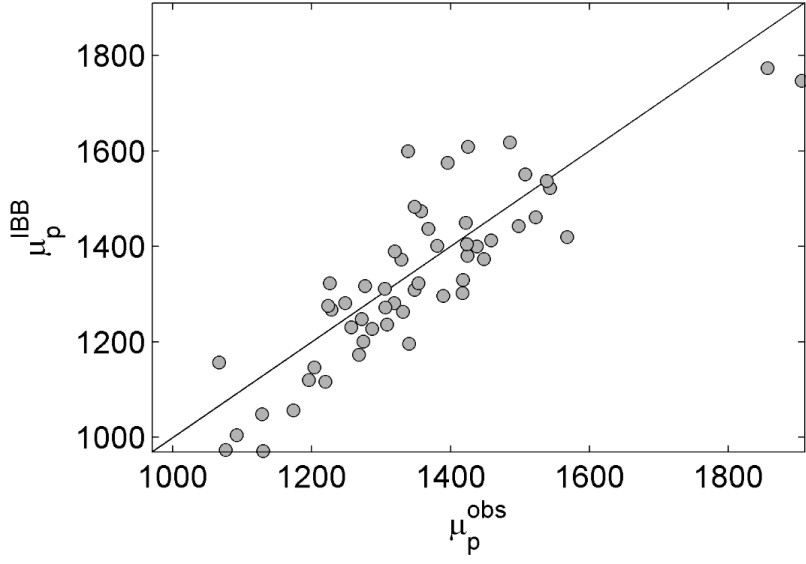

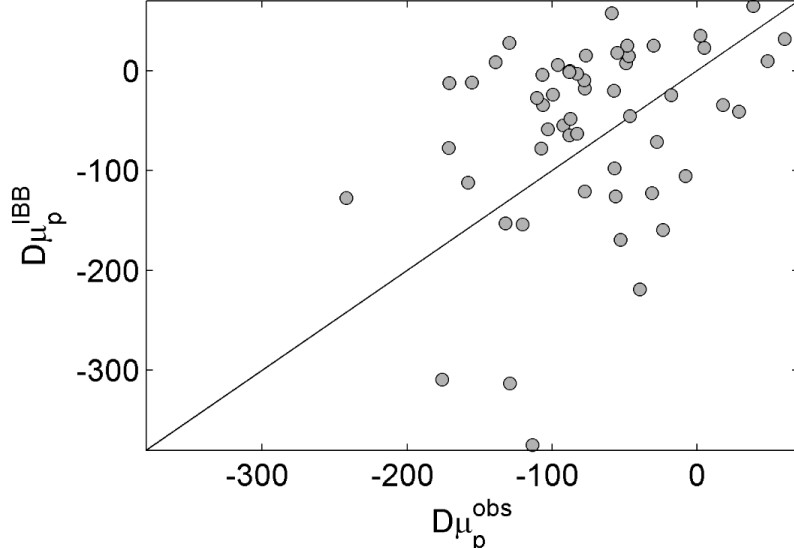

Figure 5. Annual mean precipitation (top panel) during the validation period (1994-2008) and its
difference (bottom panel) with the test period (1976-1993) for the observed data (abscissa) and
the IBB simulated data (ordinate) over all the employed stations in South Korea. For more details
about the difference at the bottom panel, see Eqs. (12) and (13).






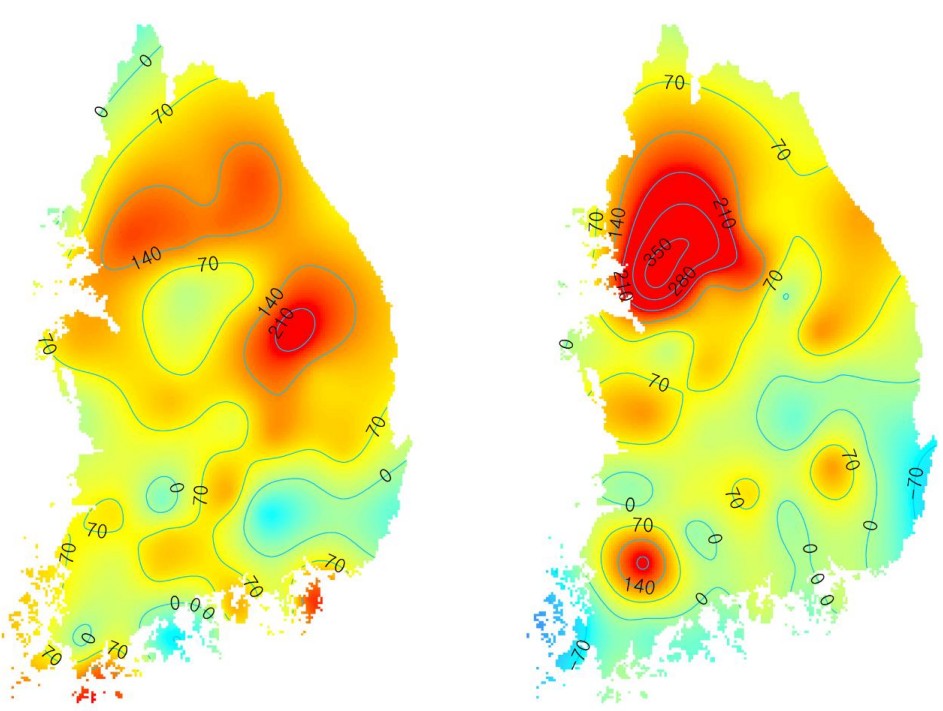


Figure 6. Spatial distributions of annual mean precipitation difference between the validation
period (1994-2008) and the test period (1976-1993) for the observed data (left panel) and the
IBB simulated data (right panel).


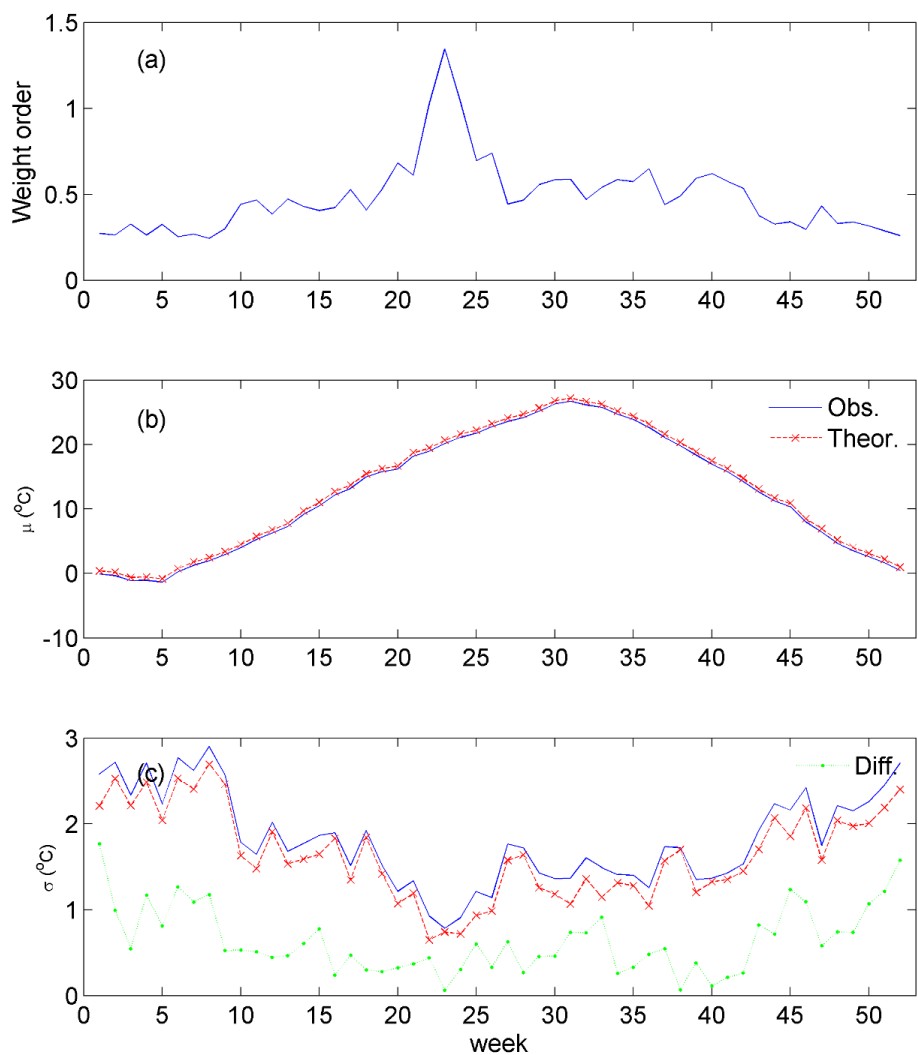


Figure 7. (a) Estimated weight order from HS and weekly statistics of (b) mean and (c) variance
for the observed temperature data (solid line) and the theoretical statistics (dashed line with cross)
using Eqs. (2) and (7) for Buan station. The weekly difference in variance between observation
and theoretical (see Eq. (8)) is shown in panel (c) by a dotted line.



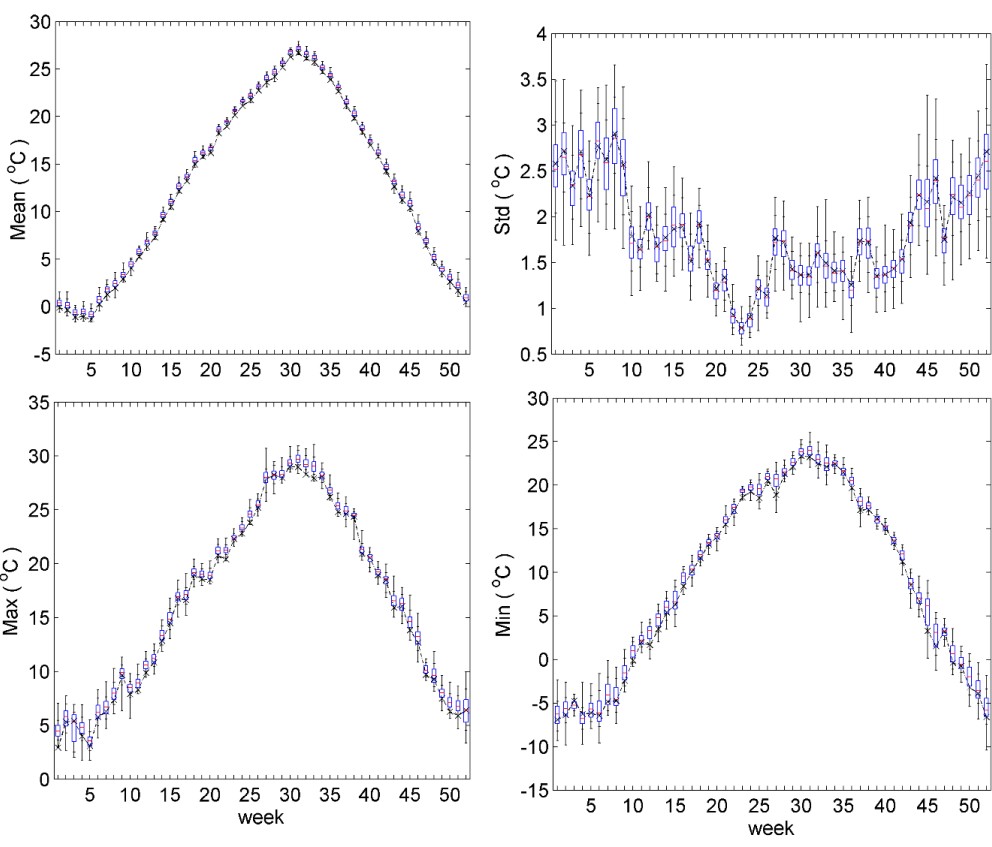


Figure 8. The statistics of the observed (dotted line with cross) and generated (boxplot) data for the weekly mean temperature using IBB with a 0.5°C temperature increase in Buan, South Korea. Boxes display the interquartile range (IQR), and whiskers extend to the extrema (i.e., maximum and minimum). The horizontal lines inside the boxes depict the median of the data. Note that the mean and maximum of the simulated data are increased significantly compared with the corresponding observed data, while the minimum of the simulated data is slightly increased and the standard deviation of the simulated data agrees with that of the observed data due to the variance inflation, as in Eq. (9).




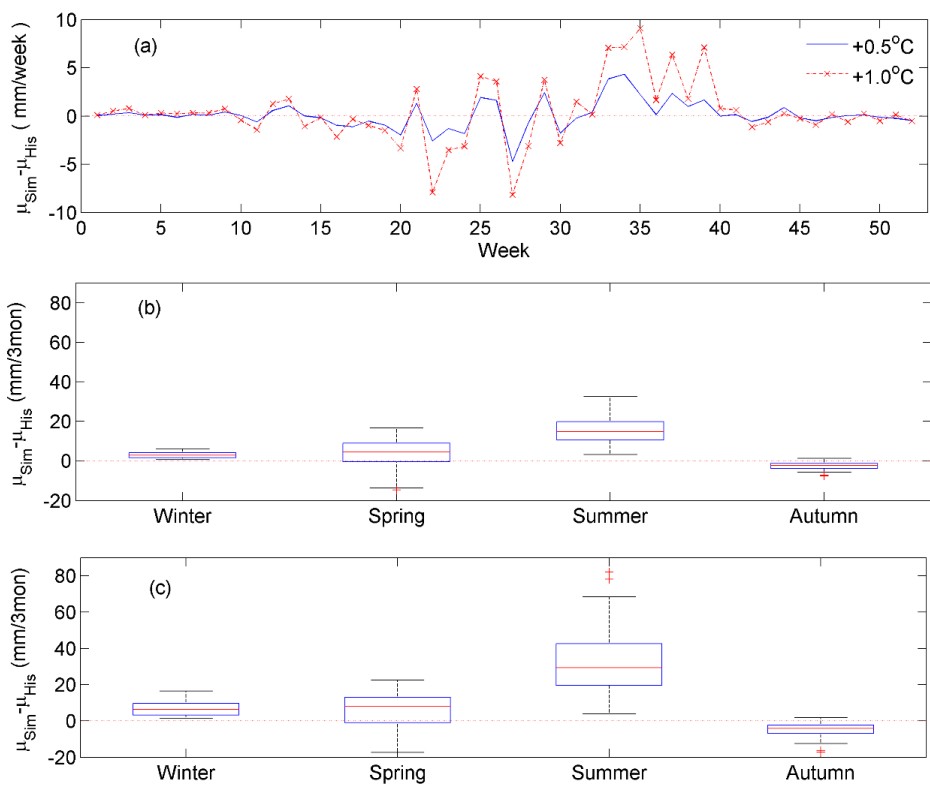

Figure 9. The mean precipitation differences of the observed and simulated data (a) for the weekly
precipitation in Buan with a 0.5°C mean temperature increase, (b) for the seasonal precipitation of
all 54 stations with a 0.5°C mean temperature increase and (c) for a 1.0°C mean temperature
increase. Note that indicates the mean of the simulated precipitation data for weekly (a) or seasonal
(b and c).






(a) Winter                    (b) Spring


(c) Summer                    (d) Autumn

Figure 10. Spatial distributions in South Korea of the mean difference in seasonal precipitation
(mm) with a 1.0℃ increase in mean temperature. Note that the scale for the summer distribution
is different from the other seasons, the 95% significance intervals are different at each station and
the mean values of the significance intervals are ±5.38, ±15.04, ±29.94, and ±4.84 for Winter
(December, January, February), Spring (March, April, May), Summer (June, July, August), and
Autumn (September, October, November), respectively.






               (a)Winter               (b)Spring



(c)Summer             (d)Autumn

Figure 11. Spatial distribution of mean difference of seasonal precipitation (mm) with 0.5ºC
increasing mean temperature in South Korea. Note that the scale of summer is different from the
other seasons and the 95% significance intervals are different at each station and the mean values
of the significance intervals are ±5.38, ±15.04, ±29.94, and ±4.84 for Winter (December, January,
February), Spring (March, April, May), Summer (June, July, August), and Autumn (September,
October, November) respectively.