# Peer review of "Simulating climate warming scenarios with intentionally biased"

_Geoscientific Model Development, 2016_

## Short Comment (SC1) · 13 Sep 2016

Dear author,

In my role as Executive editor of GMD, I would like to bring to your attention our Editorial version 1.1:

http://www.geosci-model-dev.net/8/3487/2015/gmd-8-3487-2015.html

This highlights some requirements of papers published in GMD, which is also available on the GMD website in the 'Manuscript Types' section:

http://www.geoscientific-model-development.net/submission/manuscript_types.html

In particular, please note that for your paper, the following requirements have not been met in the Discussions paper:

- "The main paper must give the model name and version number (or other unique identifier) in the title."

- "If the model development relates to a single model then the model name and the version number must be included in the title of the paper. If the main intention of an article is to make a general (i.e. model independent) statement about the usefulness of a new development, but the usefulness is shown with the help of one specific model, the model name and version number must be stated in the title. The title could have a form such as, "Title outlining amazing generic advance: a case study with Model XXX (version Y)"."

In order to simplify reference to your developments, please add a model name (or its acronym) and a version number in the title of your article in your revised submission to GMD.

Yours,

Astrid Kerkweg

---

## Referee Comment (RC1) · M. A. Ben Alaya (Referee) · 21 Nov 2016

In this paper, the author presents a statistical non parametric resampling approach called intentionally biased bootstrapping (IBB) to simultaneously simulate temperature and precipitation at a single site taking into account the increase of the temperature according to observed global warming data. The manuscript is well organized and the methodology is adequate, reasonable and clearly presented. The problematic and the application are of great interest for GMD. Hence I suggest to publish this paper. However, there are a few statements that don't entirely ring true, and I'd like the author to address these a bit more carefully. Also, drawbacks of the proposed method should be mentioned and discussed.

[Figure]

Below I list relatively minor points that could be addressed with some small revisions to the text and a few more figures:

1- Line 31: "The temperature variable is the most reliable of the GCM outputs". I'm not sure that this statement is true.

2- Line 57: I agree that moisture availability increases at the same rate with warming through the Clausius-Clapeyron (C-C) relation. Nevertheless this does not guarantee that precipitation intensity should also increase at the same rate, this presumably assumes stationarity of precipitation efficiency.

3- The proposed approach is based on the assumption that only the mean of observed temperature changes in the future, and assumes a static variance in the future. This assumption should be mentioned. Indeed the proper reproduction of the temporal variability is a very important issue, because a poor representation of the temporal variability could leads to a poor representation of extreme events.

4- Line 166: "Unlike for the case of temperature, there is no variance reduction in the resampled precipitation data because the precipitation data are not conditionally resampled"; I'm not sure that this statement is true. The existence of dependence between precipitation and temperature which motivates this work implies the existence of a concordance in the ranks of these variables. In the case of dependence there will always be some reduction in the variability of precipitation using the IBB technique. I ask the author to verify this fact by comparing the observed variance and the simulated one in the case of precipitation.

5- The proposed approach is not appropriate to simulate change in extreme events, indeed as it is the case for most resampling approach the IBB technique suffers from the inability to simulate values that are more extreme than those observed.

---

## Referee Comment (RC2) · D. Defrance (Referee) · 27 Nov 2016

This article presents a statistical method to determine local climate change from global observations. With this approach, the Intentionaly Biased Bootstrapping (IBB) and some hypothesis, the author estimates the future temperature and precipitation at a local point. The article is clearly divided into several parts: a good description of the method, the complete procedure to permit to everyone to use easily it and a good application on the South Korea to validate the method with a good description of the results. The methodology is precisely described but some information will permit to improve the comprehension. I suggest to publish this article in GMD with minor revision. The different remarks and suggestions are described below.

Some questions

Line 31: To specify that the temperature from GCM is relatively accurate as you mention in the conclusion

Line 54: In some places, such as the Sahel, the increasing in temperature results from global warming but also from feedback related to the reduction of precipitations. It is perhaps too generalist to assert that everywhere the increasing in temperature will be followed by an increasing in precipitation with the self-order of magnitude. Can this depend on the type of precipitation or the origin (e.g. monsoon system or stratiform precipitation) ?

Line 78: In the methodology, some hypothesis must be mentioned: - The method is only based on the temperature mean. If in the future the extremes of temperature increases (warmest and coldest), the method does not take this into consideration. - For the precipitation, the evolution is in relation with only the temperature evolution in the methodology and the meso-scale change is not supported.

Line 160: for the block bootstrapping technique to simulate the temperature, I would like a better description of the method with one or two sentences because it is easier to read the entire method rather than reading into the references.

Line 191: Data description, you describe the available data (74 locations) and you give 1283 mm a year but you select 54 datasets with a good hypothesis ( > 30 years available data). Is the precipitation mean the same with the only 54 datasets? I suggest to insert directly the selected datasets in the beginning of the paragraph with the hypothesis and the annual mean.

Line 250: you very accurately write that the test period is relatively short and not enough of high values of annual temperature. Did you tested a longer test period with a short validation period e. g. 20 years test period 1976-1997 and validation period 1998-2008 ?
Line 335: In the conclusion, the limits of the method in terms of variability of extremes should be recreated. This limit associated with IBB can still be disturbing for some applications such as extreme floods.

Figure 3 and 4, there are many data on it and it is not easy to analyse it for the reader. Maybe to classify the stations by order of error could permit to better interpret the results. I am not a good example to suggest to you a good representation of the results.

Technical notes

Line 58: 1 hour intensity Line 64: for this paragraph, a reference could be appreciated Line 98: local linear smoothing (Cai, 2001) Line 208: but employed in comparison ? Can you use validation ?

---

## Author Comment (AC1) · 17 Dec 2016

Author response to the reviews of the paper "Simulating warming climate scenarios with intentionally biased bootstrapping and its implications for precipitation" (Manuscript # gmd-2016-188) Taesam Lee

Line 53. White space missing Reply: The space is added accordingly.

Line 58 Clausis-Clapeyron
 -> Clausius Clapeyron Reply: The word has been modified accordingly as Clausius-Clapeyron. Note that the author do not remove '–' mark between two name because this relation has been popularly employed as is.

Line 152 Sigma2 ïĂ¡ E(Xˆ2) ïĂ▪ (EX )ˆ2 Shouldn't it be Sigma2 ïĂ¡ E(X2 ) ïĂ▪ E(X )ˆ2 ?

Please check Reply: The author checked the equation once again and found no error in the equation. It is right with $E(X^2)$.

Figure 3 legend caption. What are the red crosses? Please explain how are defined the boxes, the whiskers, the line in the boxes. Reply: The caption has been modified accordingly with adding the sentence as below. "Boxes indicate the interquartile range (IQR), and whiskers extend to +/-1.5IQR. The horizontal lines inside the boxes depict the median of the data. Data beyond the fences (+/-1.5IQR) are indicated by a plus symbol (+), which represent outliers."

Figure 4 legend caption. Please explain how are defined the boxes, the whiskers, the line in the boxes. Reply: The caption has been modified accordingly with adding the sentence as below: "Boxes indicate the interquartile range (IQR), and whiskers extend to +/-1.5IQR. The horizontal lines inside the boxes depict the median of the data. Data beyond the fences (+/-1.5IQR) are indicated by a plus symbol (+), which represent outliers."

Please also note the supplement to this comment:
http://www.geosci-model-dev-discuss.net/gmd-2016-188/gmd-2016-188-AC1-supplement.pdf

---

## Author Comment (AC2) · 17 Dec 2016

Author response to the reviews of the paper "Simulating warming climate scenarios with intentionally biased bootstrapping and its implications for precipitation" (Manuscript # gmd-2016-188) Taesam Lee Reviewer #1

M. A. Ben Alaya (Referee) In this paper, the author presents a statistical non parametric resampling approach called intentionally biased bootstrapping (IBB) to simultaneously simulate temperature and precipitation at a single site taking into account the increase of the temperature according to observed global warming data. The manuscript is well organized and the methodology is adequate, reasonable and clearly presented. The problematic and the application are of great interest for GMD. Hence I suggest to

publish this paper. However, there are a few statements that don't entirely ring true, and I'd like the author to address these a bit more carefully. Also, drawbacks of the proposed method should be mentioned and discussed. Reply: The author appreciates this reviewer's generous comment. The author tried his best efforts to improve the manuscript. Hope this improvement is satisfactory to this reviewer.

Below I list relatively minor points that could be addressed with some small revisions to the text and a few more figures: 1- Line 31: "The temperature variable is the most reliable of the GCM outputs". I'm not sure that this statement is true. Reply: The author really appreciates this reviewer's detailed comment. The sentence was modified accordingly as: "The temperature variable is more reliable than other variables in GCM outputs."

2- Line 57: I agree that moisture availability increases at the same rate with warming through the Clausius-Clapeyron (C-C) relation. Nevertheless this does not guarantee that precipitation intensity should also increase at the same rate, this presumably assumes stationarity of precipitation efficiency. Reply: The author totally agrees with this reviewer's comment. The sentence was circumvented as follows: " From the Clausius-Clapeyron (C-C) relation, saturation vapor pressure increases by 6-7% for each 1oC increase in temperature and rainfall intensity also increases in a similar rate with warming (Trenberth and Shea, 2005)." 3- The proposed approach is based on the assumption that only the mean of observed temperature changes in the future, and assumes a static variance in the future. This assumption should be mentioned. Indeed the proper reproduction of the temporal variability is a very important issue, because a poor representation of the temporal variability could leads to a poor representation of extreme events. Reply: The author really appreciates this reviewer's insightful comment. The limitation and its possible development is discussed at the conclusion section as the below. Hope this modification is satisfactory to this reviewer.

"The proposed IBB method is conditioned and assumed only on the mean temperature change. A further scheme can be developed to consider the changes of multiple

variables with classifying the conditions of interested variable."

For the relation of the temporal variability and extreme events, the author consider that this reviewer's comment can be true but not always as far as this reviewer's viewpoint. Further study relates on this issue can be studied in near future.

4- Line 166: "Unlike for the case of temperature, there is no variance reduction in the resampled precipitation data because the precipitation data are not conditionally resampled"; I'm not sure that this statement is true. The existence of dependence between precipitation and temperature which motivates this work implies the existence of a concordance in the ranks of these variables. In the case of dependence there will always be some reduction in the variability of precipitation using the IBB technique. I ask the author to verify this fact by comparing the observed variance and the simulated one in the case of precipitation. Reply: The author really appreciates this reviewer's insightful comment. The author compared the observed variance with the simulated one for all the 54 stations. No significant variance reduction was observed and even some stations (15 stations) present variance inflation (i.e. simulated variance is bigger than the observed variance). Therefore, the author consider that the statement can be true but with a little less certainty. The sentence was modified as: "not much significant variance reduction is expected in the resampled precipitation data because the precipitation data are not conditionally resampled."

5- The proposed approach is not appropriate to simulate change in extreme events, indeed as it is the case for most resampling approach the IBB technique suffers from the inability to simulate values that are more extreme than those observed. Reply: The author really appreciates this reviewer's insightful comment. The authors consider that long-term variability of extremes can be derived from the IBB method when it is related with other variables such as precipitation. But it might be limited since no physical mechanisms can be included. This limitation and possible extension were discussed from this reviewer and the other reviewer's comment at the conclusion as follows:

"The proposed IBB method is not a physical-based method but a statistical simulation approach in which a physical mechanism of precipitation cannot be taken into consideration. Substantial modification might be required to accommodate this mechanism. The proposed IBB method is conditioned and assumed only on the mean temperature change. A further scheme can be developed to consider the changes of multiple variables with classifying the conditions of interested variable. Another possible extension of the current study must be on analyzing the future variation of hydrological extreme events (e.g. extreme floods). When a long-term variation of hydrological extreme events is related with precipitation, the proposed IBB method can be used to derive the variation. "

Please also note the supplement to this comment:
http://www.geosci-model-dev-discuss.net/gmd-2016-188/gmd-2016-188-AC2-supplement.pdf

―――――――――――――――

---

## Author Comment (AC3) · 17 Dec 2016

Author response to the reviews of the paper "Simulating warming climate scenarios with intentionally biased bootstrapping and its implications for precipitation" (Manuscript # gmd-2016-188) Taesam Lee Reviewer #2 D. Defrance (Referee) This article presents a statistical method to determine local climate change from global observations. With this approach, the Intentionaly Biased Bootstrapping (IBB) and some hypothesis, the author estimates the future temperature and precipitation at a local point. The article is clearly divided into several parts: a good description of the method, the complete procedure to permit to everyone to use easily it and a good application on the South Korea to validate the method with a good description of the results. The methodology

is precisely described but some information will permit to improve the comprehension. I suggest to publish this article in GMD with minor revision. The different remarks and suggestions are described below. Reply: The author appreciates this reviewer's generous comment. The author tried his best efforts to improve the manuscript. Hope this improvement is satisfactory to this reviewer.

Some questions Line 31: To specify that the temperature from GCM is relatively accurate as you mention in the conclusion Line 54: In some places, such as the Sahel, the increasing in temperature results from global warming but also from feedback related to the reduction of precipitations. It is perhaps too generalist to assert that everywhere the increasing in temperature will be followed by an increasing in precipitation with the self-order of magnitude. Can this depend on the type of precipitation or the origin (e.g. monsoon system or stratiform precipitation) ? Reply: The author appreciate this reviewer's detailed comment. The proposed IBB method does not postulate that the temperature increase implies the increase of precipitation. The method employs the empirical relation between temperature and precipitation. When an observed temperature increases and an observed precipitation decreases, the same reverse relation can be reproduced through the proposed IBB method. The author considers that the proposed method is not physical-based method so that the type of precipitation cannot be taken into consideration. This limitation is mentioned at the end of the conclusion section from the comment of this reviewer below.

Line 78: In the methodology, some hypothesis must be mentioned: - The method is only based on the temperature mean. If in the future the extremes of temperature increases (warmest and coldest), the method does not take this into consideration. - For the precipitation, the evolution is in relation with only the temperature evolution in the methodology and the meso-scale change is not supported. The author really appreciates this reviewer's insightful comment. No physical mechanisms can be included. This limitation was discussed at the conclusion section.

"The proposed IBB method is not a physical-based method but a statistical simulation

approach in which a physical mechanism of precipitation cannot be taken into consideration. Substantial modification might be required to accommodate this mechanism."

Line 160: for the block bootstrapping technique to simulate the temperature, I would like a better description of the method with one or two sentences because it is easier to read the entire method rather than reading into the references. Reply: The author totally agrees with this reviewer's comment. Simple sentences were added accordingly as follows:

"Bootstrapping is a random sampling with replacement and block bootstrapping is to resample blocks. Each block contains a set of predictor and predictand like a regression. Here, temperature and precipitation can be set as a block and they act as predictor and predictand, respectively."

The author hopes that this modification is satisfactory to this reviewer

Line 191: Data description, you describe the available data (74 locations) and you give 1283 mm a year but you select 54 datasets with a good hypothesis ( > 30 years available data). Is the precipitation mean the same with the only 54 datasets? I suggest to insert directly the selected datasets in the beginning of the paragraph with the hypothesis and the annual mean. Reply: The author appreciate this reviewer's detailed comment. Official annual mean precipitation of South Korea (1283mm) is announced by KMA, not calculated from the current study. The sentence was modified accordingly as follows:

"In the current study, 54 weather stations that record temperature and precipitation in South Korea with more than 30 years of record length and that are managed by the Korea Meteorological Administration (KMA) were employed. South Korea is located in Far East Asia and has a mean annual precipitation of 1283 mm from KMA."

The author hopes that this modification is satisfactory to this reviewer

Line 250: you very accurately write that the test period is relatively short and not

enough of high values of annual temperature. Did you tested a longer test period with a short validation period e. g. 20 years test period 1976-1997 and validation period 1998-2008 ? Reply: The author really appreciates this reviewer's pinpointing comment. 20 years was also tested with no difference from the current test. 15 years (the test period that has been used in the current study) and 20 years are not much different from 15 years in analyzing the long-term change.

Line 335: In the conclusion, the limits of the method in terms of variability of extremes should be recreated. This limit associated with IBB can still be disturbing for some applications such as extreme floods. Figure 3 and 4, there are many data on it and it is not easy to analyse it for the reader. Maybe to classify the stations by order of error could permit to better interpret the results. I am not a good example to suggest to you a good representation of the results. Reply: The author really appreciates this reviewer's insightful comment. The authors consider that long-term variability of hydrological extremes can be derived from the IBB method when it is related with other variables such as precipitation. But no physical mechanisms can be included as this reviewer pointed in the previous comment. This limitation and possible extension were discussed at the conclusion as follows:

"The proposed IBB method is not a physical-based method but a statistical simulation approach in which a physical mechanism of precipitation cannot be taken into consideration. Substantial modification might be required to accommodate this mechanism. Also, a possible extension of the current study must be on analyzing the future variation of hydrological extreme events (e.g. extreme floods). If a long-term variation of hydrological extreme events is related with precipitation, one can derive the variation from the IBB method."

In Figure 3 and 4, the classification of the station by order of error is not easy since the magnitude of error varies all times and it is not good to change the order of stations every time. The temperature and precipitation behave differently for their changes at each station. Therefore, the author consider that it is better to stand the station order

as is.

Hope this reviewer satisfactory to this modification.

Technical notes Line 58: 1 hour intensity Reply: It was modified as 'the intensity of hourly precipitation'. Hope this modification is satisfactory to this reviewer.

Line 64: for this paragraph, a reference could be appreciated Reply: A reference is added accordingly.

Line 98: local linear smoothing (Cai, 2001) Reply: It was modified as 'local linear regression'.

Line 208: but employed in comparison ? Can you use validation ? Reply: The author appreciates this reviewer's detailed comment. 'validation' was used now according to this reviewer's comment.

Please also note the supplement to this comment:
http://www.geosci-model-dev-discuss.net/gmd-2016-188/gmd-2016-188-AC3-supplement.pdf